# Plasma-Assisted Preparation of Reduced Graphene Oxide and Its Applications in Energy Storage

**DOI:** 10.3390/nano14231922

**Published:** 2024-11-29

**Authors:** Haiying Li, Yufei Han, Pengyu Qiu, Yuzhe Qian

**Affiliations:** 1College of Architecture, Nanjing Tech University, Nanjing 211816, China; 2Institute of International Education, New Era University College, Kajang 43000, Malaysia; 3Faculty of Maths & Physical Sciences, University College London, London WC1E 6AE, UK; rachelhyf526@gmail.com; 4College of Electrical Engineering and Control Science, Nanjing Tech University, Nanjing 211816, China; 202221111047@njtech.edu.cn (P.Q.); aurora_jason@njtech.edu.cn (Y.Q.)

**Keywords:** reduced graphene oxide, plasma-assisted reduction, working gas, energy storage devices

## Abstract

Reduced graphene oxide (rGO) exhibits mechanical, optoelectronic, and conductive properties comparable to pristine graphene, which has led to its widespread use as a method for producing graphene-like materials in bulk. This paper reviews the characteristics of graphene oxide and the evolution of traditional reduction methods, including chemical and thermal techniques. A comparative analysis reveals that these traditional methods encounter challenges, such as toxicity and high energy consumption, while plasma reduction offers advantages like enhanced controllability, the elimination of additional reducing agents, and reduced costs. However, plasma reduction is complex and significantly influenced by process parameters. This review highlights the latest advancements in plasma technology for reducing graphene oxide, examining its effectiveness across various gas environments. Inert gas plasmas, such as argon (Ar) and helium (He), demonstrate superior reduction efficiency, while mixed gases facilitate simultaneous impurity reduction. Additionally, carbon-based gases can aid in restoring defects in graphene oxide. This paper concludes by discussing the future prospects of plasma-reduced graphene and emphasizes the importance of understanding plasma parameters to manage energy and chemical footprints for effective reduction.

## 1. Introduction

Graphene is an allotrope of carbon composed of a single layer of sp^2^-hybridized carbon atoms arranged in a two-dimensional honeycomb lattice [1]. Since its discovery in 2004, this monolayer carbon material has garnered significant attention from researchers worldwide due to its remarkable properties. These include a high surface area, excellent Young’s modulus, outstanding thermal and electrical conductivity, and good optical transparency [2,3,4,5]. Such unique physicochemical characteristics have underscored graphene’s immense potential across various applications, encompassing energy conversion and storage (including fuel cells and capacitors), sensors, electrocatalysis, and electronic devices.

The synthesis of graphene can be achieved through a variety of methods, including mechanical cleavage, epitaxial growth, chemical vapor deposition, electrochemical exfoliation of graphite [6,7,8], and the reduction of graphene oxide (GO), which is produced through the chemical exfoliation of graphite [9]. Recently, non-covalent exfoliation of graphite in liquid phases has also been reported. Among these methods, the reduction of GO is considered one of the most promising pathways for the large-scale, cost-effective production of graphene [10,11]. However, this approach only partially restores the original properties of graphene. For instance, chemically reduced rGO typically achieves an oxygen-to-carbon (O/C) ratio of 0.1–0.2, compared to pristine graphene’s O/C ratio close to 0, and an electrical conductivity in the range of 10–100 S·cm^−1^, far below the 10^4^–10^5^ S·cm^−1^ observed for pristine graphene [12,13]. Consequently, the resulting product is often referred to as reduced graphene oxide (rGO), emphasizing its distinct characteristics compared to pristine graphene. GO is known for its hydrophilicity and electrical insulating properties [14]. These characteristics arise from the disruption of the sp^2^ bond network on its carbon-based surface during the chemical exfoliation of graphite. This process leads to a significant portion of the sp^2^ carbon network bonding with oxygen-containing functional groups.

There are several reduction pathways for GO, including chemical reduction, thermal reduction, photocatalytic reduction, and electrochemical reduction [15,16,17]. Among these, chemical and thermal reductions are the most commonly used methods for large-scale industrial production. However, chemical reduction typically requires the use of toxic reducing agents, such as hydrazine, which pose environmental risks. On the other hand, thermal reduction involves high temperatures to eliminate oxygen functional groups, resulting in complex experimental conditions and higher production costs. To tackle these challenges, researchers have pursued various optimization strategies. For example, Lavin et al. reported that by integrating chemical and thermal reduction methods, GO reduction efficiency could be significantly improved under mild temperature conditions of approximately 150–200 °C [18]. Such conditions are considerably lower compared to conventional thermal reduction, which typically requires temperatures exceeding 800 °C. Meanwhile, Silva et al. concentrated on optimizing reducing agents in the chemical reduction process, exploring the potential for environmentally friendly alternatives [19]. Although photocatalytic and electrochemical reductions are relatively eco-friendly and easy to implement, they face drawbacks such as high energy consumption and low product selectivity [14,20].

In contrast, plasma reduction has emerged as a promising technique, being not only simple and cost-effective but also rapid and environmentally sustainable for reducing GO [21,22,23,24]. For example, thermal reduction at 1000 °C can achieve an oxygen content below 5%, while plasma reduction achieves a comparable level of oxygen removal at room temperature, significantly reducing energy consumption. Additionally, the electrical conductivity of plasma-reduced rGO has been reported to reach 200–500 S·cm^−1^, outperforming most chemically reduced rGO samples [25]. Researchers have successfully harnessed high-energy reactive species produced by plasma to reduce GO, yielding promising outcomes [26]. For example, Zhang et al. utilized alternating current and nanosecond power to drive argon gas in a dielectric barrier discharge, resulting in the preparation of reduced graphene oxide/nickel oxide (rGO/NiO) nanocomposites. Performance tests demonstrated a substantial reduction in oxygen functional groups [27]. Similarly, Abdelkader-Fernández et al. achieved an effective reduction of GO using hydrogen plasma, reaching a high degree of reduction even at ambient temperatures [28]. Plasma-related theories indicate that the reduction effect on GO differs based on the working gas utilized. While many existing studies concentrate on optimizing reactor parameters tailored to their specific processes, the mechanisms by which various elemental plasmas affect the reduction of GO are not yet fully understood. Consequently, it is essential to summarize and explore the current techniques for the plasma reduction of GO. By quantitatively comparing the performance of plasma reduction with traditional methods in terms of efficiency, environmental impact, and scalability, this review aims to provide insights into the future direction of GO reduction technologies.

In this article, we first introduce the properties of GO and reduced GO, followed by a detailed overview of traditional reduction methods, including their operational conditions and performance metrics. Quantitative comparisons are made to highlight the limitations of these methods, such as the high energy demand of thermal reduction or the environmental risks of chemical reduction. Subsequently, the focus shifts to plasma reduction, emphasizing its advantages in achieving significant oxygen removal and high electrical conductivity under mild conditions. This review further discusses the reduction mechanisms under varying plasma parameters, the challenges in optimizing plasma systems, and potential strategies to improve process efficiency. Finally, we explore the application of plasma-reduced rGO in energy storage devices, offering a comprehensive perspective on future research directions.

## 2. Properties of GO and Traditional Reduction Methods

### 2.1. Structure and Properties of GO

Understanding the structure and properties of GO is crucial for clarifying the reduction process. GO is typically described as a material made up of single-layer graphene sheets, characterized by the presence of hydroxyl (-OH), carboxyl (-COOH), and epoxy (-O-) functional groups on both its basal plane and edges. This configuration leads to a mixture of sp^2^- and sp^3^-hybridized carbon atoms [29,30,31]. Such hybridization not only affects the material’s chemical reactivity but also imparts exceptional mechanical properties and electrical conductivity. The diversity and complexity of GO give it significant potential for applications in areas such as energy storage, catalysis, and composite materials [32,33]. Over the years, researchers have utilized a range of advanced instrumental techniques to conduct comprehensive analyses of the structure of GO [34]. These techniques include annular dark field imaging, ultra-high-resolution transmission electron microscopy, and X-ray diffraction (XRD) [35,36]. Despite extensive efforts to clarify the microscopic structure of GO, multiple structural models continue to exist, and achieving a broad consensus remains challenging [36,37,38]. This ongoing variability can primarily be attributed to the material’s inherent complexity and the unique stoichiometric properties of each sample, resulting in diverse research findings and associated uncertainties.

As of now, considerable uncertainty persists regarding the precise chemical structure of GO [39,40,41]. The most widely accepted model, proposed by Lerf and Klinowski (LK) in 1996, distinguishes between two distinct types of regions: the hexagonal aliphatic ring region and the non-oxidized aromatic ring region, as shown in Figure 1 [42,43]. The dimensions of these regions depend on the degree of oxidation of the material. This model primarily consists of aromatic units, epoxy groups, and double bonds. The wrinkling observed in the monolayer is attributed to subtle distortions in the tetrahedral geometry of the hydroxyl groups associated with the carbon atoms. Oxygen functional groups are attached to both the upper and lower surfaces of the monolayer carbon, resulting in two layers of oxygen atoms with varying concentrations, primarily composed of epoxides and hydroxyls that are in close proximity. All oxygen functional groups, aromatic units, and oxidized rings are distributed randomly throughout the carbon monolayer. The acidity of GO can be explained by the presence of oxygen groups, such as hydroxyl (-OH) and carboxyl (-COOH), which are affixed to the edges of the lattice.

GO is defined by the presence of oxygen-containing functional groups on its surface. These polar groups make GO hydrophilic, which facilitates its dispersion in water [44]. However, the oxygen functional groups at the edges also increase the interlayer spacing of GO, resulting in a heterogeneous chemical and electronic structure that limits its electrical and thermal conductivity [36]. In a complex oxidative environment, the ordered sp^2^ regions on the carbon basal plane are disrupted by these oxygen functional groups, leading to the formation of disordered sp^3^ regions at the edges. Shreya et al. used transmission electron microscopy (TEM) to examine the defect structure of GO [31], as shown in Figure 2. GO has a structure that consists of long-range ordered sp^2^ regions, with small patches of amorphous material or defects at the edges. Therefore, it is crucial to consider both the efficiency of oxygen group removal and the inherent structure of GO during the reduction process. While a higher rate of oxygen removal can greatly improve the electrical conductivity of the final product, the reduction process may also disturb the ordered lattice, which could hinder its conductivity. Additionally, by controlling the ratio and distribution of sp^2^ and sp^3^ regions through reduction, the optical properties of rGO can be improved [45]. The reduction in defects and impurities can lead to more efficient electron–hole recombination, enhancing the emission of blue light.

### 2.2. The Reduction Methods of GO

The irregular and random defects, along with oxygen-containing groups in the GO structure, restrict its electrical and thermal performance. As a result, reduction is necessary to obtain rGO, which has a broader range of applications. The existing reduction methods include chemical reduction [46], thermal reduction [47], and photocatalytic reduction [48], with chemical and thermal reductions being the primary methods currently used in industry.

Chemical reduction of GO has been documented for over 50 years, primarily leveraging the strong reducing and nucleophilic properties of various reducing agents to facilitate the opening of oxygen-containing groups for reduction. In 2007, the Ruoff team introduced the concept of synthesizing graphene through the reduction of GO, utilizing hydrazine as the reducing agent [49]. This method is straightforward and achieves a high degree of reduction; however, it often results in the aggregation of the resultant graphene, significantly limiting its applications. Alkaline compounds, such as NaOH and KOH, similar to hydrazine, can also be employed for GO reduction [50], offering advantages such as non-toxicity and operational simplicity. As the concentration of these alkaline solutions increases, reduction reactions can occur at room temperature, though the resulting reduction efficiency may be suboptimal.

In recent years, novel reducing agents have been developed, including amine, sulfur-containing, and hydroxyl-containing compounds, each exhibiting certain limitations [51,52,53]. For instance, research by Mellon et al. identified that thiol groups (-SH) possess both reducing capabilities and strong nucleophilicity, enabling them to reduce GO through nucleophilic addition reactions [54]. This approach, while straightforward and effective in incorporating sulfur into the product to enhance conductivity, can produce byproducts that have unpleasant odors and potential toxicity. Tran et al. explored the use of non-aromatic, sulfur-free amino acids as reducing agents for GO [55], successfully producing rGO nanosheets through an environmentally friendly process suitable for large-scale applications. Similarly, Silva et al. demonstrated the reduction of GO using ascorbic acid as a green reducing agent [56], as shown in Figure 3, which is notable for its strong reducing capacity, safety, and low cost, making it one of the most widely employed reduction methods today. Additionally, some researchers have investigated the use of biomaterials for reduction [57,58], primarily relying on fundamental substances such as amino acids and vitamins found in biomass. However, the resultant products may contain impurities that necessitate further purification. In conclusion, current research on chemical reduction methods emphasizes achieving environmentally friendly processes while maintaining high reduction efficiency. The selection of reducing agents must also align with practical industrial requirements in real-world applications.

Electrochemical reduction is another promising method for reducing GO, involving the application of an electrical potential to a GO-coated electrode in an electrolyte solution [59]. This potential induces electron transfer to the oxygen-containing groups, reducing them to water or volatile compounds while partially restoring the sp^2^ carbon network [60]. Unlike chemical reduction, which often requires toxic agents, or thermal reduction, which consumes high energy, electrochemical reduction operates under mild conditions, typically at room temperature. It offers the advantages of being environmentally friendly and allowing precise control over the reduction process. Chernova et al. prepared proton-selective membranes via electrochemical reduction of GO, with the C/O ratio of the resulting product ranging from 2 to 4 [61]. However, its efficiency is generally lower than that of thermal methods, resulting in reduced conductivity compared to thermally reduced rGO. Despite these limitations, electrochemical reduction is well suited for applications requiring preserved functional groups, such as in biosensors and flexible electronics [62].

Thermal reduction is a widely used method for reducing GO. The principle of this approach involves the thermal decomposition or evaporation of oxygen functional groups (such as hydroxyl, ether, and carboxyl groups) at elevated temperatures, which helps restore some of the sp^2^ carbon structure [63]. Proper thermal reduction supports the preservation and transformation of oxygen-containing groups and structural defects, promoting the formation of reactive carbon radicals and discrete graphite domains. This enhancement facilitates the activation of molecular oxygen and near-plasma photothermal effects [64], thereby restoring the conductive network [65]. Moreover, thermal reduction can also be employed in the synthesis of graphene-based materials and devices through processes such as thermal annealing [66] and hydrothermal methods [67]. By optimizing these processes, it is possible to address the cost issues associated with excessively high temperatures.

A review of commonly used methods for reducing GO indicates that chemical reduction has reached a relatively mature stage of development, with current efforts focusing on achieving environmentally friendly processes while striving for higher reduction efficiency. In contrast, thermal reduction, based on a relatively straightforward principle, has seen researchers increasingly emphasize enhancing the electrical and optical properties of the resulting products while maintaining consistent processing outcomes. However, both methods have inherent limitations. For example, chemical reduction consumes reducing agents, and the pursuit of high reduction rates may introduce impurities into the final product. Conversely, thermal reduction is associated with higher costs and does not effectively address defects present in GO.

Plasma reduction has emerged as a promising new method for reducing GO [68]. It generates high-energy reactive species to remove oxygen functional groups, and depending on the working gas employed, it can eliminate impurities incorporated within GO and repair defects on the carbon basal planes [69]. Additionally, this method allows for efficient doping tailored to specific application requirements, making it a hot topic in current research and development [70].

During the plasma discharge process, gases such as hydrogen, nitrogen, or argon are exposed to a high-energy electric field, which ionizes them and generates high-energy electrons, ions, free radicals, and reactive molecules [71]. These reactive species possess sufficient energy to break the chemical bonds on the surface of GO. Taking hydrogen plasma as an example, high-energy hydrogen radicals and molecules react with the oxygen-containing functional groups on GO, producing small molecules like H_2_O, CO, and CO_2_. These small molecules then evaporate from the surface, thereby reducing the oxygen content in the GO. High-energy electrons in the plasma can also directly bombard the GO surface, disrupting carbon–oxygen bonds, creating carbon vacancies, or restoring the aromatic structure [72]. Unlike thermal or chemical reduction, the high-energy characteristics of plasma enable efficient reduction at lower overall temperatures, while minimizing damage to the graphene lattice structure [73].

## 3. Plasma-Assisted Reduction of GO

Atmospheric pressure plasma is generated through high-voltage ionization, which results in a high concentration of electrons, ions, and reactive free radicals [74]. This process also produces ultraviolet radiation, heat, and electromagnetic fields that impact material preparation, affecting both their physical structure and chemical composition [75]. The plasma-assisted reduction of GO can be compared to plasma etching processes [76], as it selectively removes oxygen-containing groups while preserving the underlying carbon network of GO. This selective removal of oxygen-containing groups makes the plasma reduction process particularly efficient for reducing GO. During plasma exposure, two processes occur simultaneously on the solid surface: (1) material deposition and (2) ablation leading to material removal. Material deposition involves reactive species in the plasma, such as ions, radicals, or neutral atoms, interacting with the GO surface to transform oxygen-containing groups into gaseous byproducts (e.g., CO or CO_2_), while also potentially introducing new functional structures on the surface. Ablation, on the other hand, selectively breaks the chemical bonds of oxygen-containing groups through high-energy particle collisions or photon irradiation, detaching them from the GO surface. These two processes complement each other, enabling the efficient removal of oxygen-containing groups while preserving the ordered carbon network of GO [77]. This ensures that the rGO retains excellent electrical conductivity and a well-maintained crystal structure, thereby enhancing its applicability in electronics, energy, and catalysis. In recent years, numerous researchers have successfully controlled plasma performance by adjusting discharge parameters [78], thereby optimizing the reduction of GO. Among these parameters, the choice of working gases—such as inert gases (argon, nitrogen) and reducing gases (ammonia, hydrogen)—plays a significant role in the plasma process. Different types of gas can substantially influence plasma’s electron temperature and density, which in turn affects its reactivity and energy transfer efficiency [79,80]. Furthermore, the chemical properties of the gases may interact with the material, modifying the physical and chemical characteristics of the final product. This section summarizes the mechanisms by which various working gases impact the plasma reduction of GO and discusses the development of related processes.

### 3.1. Inert Gases

Inert gases exhibit low chemical reactivity and cannot provide reducing agents for chemical reactions. However, when high voltage is applied to a reactor filled with inert gas, a significant number of high-energy electrons and ions are generated within the discharge gap. Under the influence of a strong electric field, these high-energy species collide with deformed chemical bonds, resulting in bond cleavage [81]. This rapid bond-breaking produces substantial amounts of H_2_O and CO_2_, facilitating the reduction of GO. Additionally, high-temperature conditions can promote the pyrolytic reactions of oxygen-containing functional groups in GO [82,83]. The energy within the plasma reactor not only affects the material but also causes a sharp increase in the temperature of the discharge gap. Therefore, under the synergistic effects of high energy and high temperature, inert gas plasmas exhibit excellent reducing capabilities.

Sui et al. employed argon non-thermal radio frequency plasma technology to reduce inkjet-printed GO films, successfully converting electrically insulating GO into conductive rGO [84]. This method offers the advantage of operating at moderate temperatures, thereby avoiding the performance degradation issues associated with high temperatures in traditional thermal reduction methods. Furthermore, this plasma process is compatible with a variety of substrates, including electrically insulating, temperature-sensitive, and absorbent materials such as photo paper, thus expanding the potential applications of rGO in flexible electronic devices. As shown in Figure 4, Fourier-transform infrared spectroscopy (FTIR) analysis showed that the effectiveness of plasma reduction is comparable to that of electrochemical reduction, while its reduction efficiency is higher than that of thermal reduction methods, highlighting significant advantages for practical applications.

Mohai et al. successfully reduced the oxygen content of GO films from 29% to 21% using argon plasma treatment, which significantly enhanced their conductivity [85]. This achievement not only improved the conductivity of GO but also highlighted the vital role of plasma technology in modifying the chemical properties of material surfaces. Meanwhile, Ibrahim et al. explored reduced GO films deposited using the Langmuir–Blodgett (LB) method, examining the effects of argon plasma treatment on surface optimization and structural characteristics [86]. After treatment, the wettability of the GO films improved significantly, and their adhesion to glass substrates was strengthened. These enhancements contribute to the stability and practicality of films for various applications. Furthermore, cyclic voltammetry tests indicated that the treated films displayed superior electrochemical performance.

Kim et al. [87] achieved selective etching of monolayer graphene using an inductively coupled plasma ion beam system, successfully avoiding damage to the underlying substrate. This advancement opens new possibilities for the application of graphene in electronic devices. The study introduced a cyclic etching process that initially treats the surface with chemically adsorbed low-energy oxygen ions (O_2_^+^ and O^+^, with an energy range of 0–20 eV), followed by physical desorption of the oxidized material using Ar^+^ ions (with an energy of 11.2 eV). The innovation of this method lies in the design of a floating gate and grounded gate with an axial magnetic field, which effectively optimizes the ion energy distribution under varying power and gas flow conditions, allowing for precise etching of the top graphene layer. Notably, the study revealed that the binding energy of surface carbon atoms significantly decreased during the chemical adsorption of oxygen ions, dropping from approximately 6.1 eV to about 3.9 eV, while the binding energy of the underlying C-C bonds remained relatively unchanged (around 0.1 eV). This observation provides a theoretical basis for selective etching.

Additionally, the synergistic effect of controlled energy from Ar^+^ ions and oxygen ions impart high selectivity and controllability to the etching process. This research paves the way for the effective and controlled reduction of GO by leveraging the combined chemical and energetic influences of plasma species.

### 3.2. Hydrogen

A molecular dynamics study on graphene indicates that hydrogen atoms with energies between 0.025 and 0.3 eV can selectively etch the edges without damaging the basal plane, whereas hydrogen atoms with energies ranging from 0.3 to 10 eV can hydrogenate the basal plane, irreversibly damaging the graphene structure [88]. Dielectric barrier discharge (DBD) plasma generates high electron temperatures (1–10 eV) and high electron densities (10^18^–10^21^ m^−3^), making it capable of breaking chemical bonds with dissociation energies below 10 eV [89]. Hydrogen plasma initially produces a substantial amount of reactive hydrogen radicals and electrons, which continuously bombard the chemical bonds of GO. The oxygen-containing groups are destroyed and removed under sustained bombardment by high-energy species, resulting in the rapid generation of H_2_O and CO_2_ within nanoseconds. This process creates high pressure between the layers of GO, leading to its reduction and delamination.

Li et al. [90] developed an innovative method for boron-doped reduction of GO using hydrogen gas as the working gas in a coaxial DBD reactor. Comparative characterization of the samples before and after treatment using XRD and FTIR demonstrated the effective removal of oxygen-containing groups from the precursor, significantly enhancing the reduction of GO, as shown in Figure 5. The appearance of the 002 peak in the XRD spectrum further confirmed the reduction of the GO structure. This method, utilizing hydrogen as a reducing atmosphere, not only provides an effective pathway to decrease the oxygen content in GO but also creates a favorable environment for boron doping, potentially enhancing the material’s conductivity. Moreover, the design of the coaxial DBD reactor optimized gas flow and reaction conditions, making the process more efficient and controllable.

Li et al. [91] investigated how different plasma powers and Ar/H_2_ gas mixtures affect the reduction of GO, highlighting the potential applications of plasma chemistry in materials preparation. In their experiments, the authors found that the emission intensity of atomic hydrogen increased with higher discharge power, reaching its peak at a H_2_/Ar ratio of 2:1, as shown in Figure 6. This phenomenon is likely due to the encapsulation effect of Ar, which promotes the Penning ionization of H_2_ molecules. This process accelerates hydrogen dissociation and enhances the efficiency of the reduction reaction. The experimental results showed significant variations in the C/O ratio of the rGO under different gas conditions. Using pure Ar and pure H_2_ plasmas for the reduction resulted in C/O ratio values of 1.2 and 1.7, respectively, while the H_2_/Ar (2:1) condition produced a C/O ratio value as high as 6.9. This indicates that optimizing the gas mixture ratio is essential for improving reduction efficiency. Moreover, the study demonstrated that the rGO exhibited excellent electrochemical performance as an electrode material, achieving a specific capacitance of 185.2 F/g in tests using a KOH aqueous solution.

### 3.3. Methane

To obtain high-quality graphene samples using GO as a precursor, numerous studies on defect healing (or repair) reduction have been conducted. Methane plasma is a commonly used method for repairing defects in GO and reducing GO [92,93], as it not only provides a hydrogen source to remove oxygen-containing groups but also generates a carbon source to repair the defects caused by the removal of these groups.

Morikuni et al. [94] prepared monolayer GO films using a spin-coating method and effectively reduced these films in a short time using methane plasma, demonstrating the innovation and advantages of this approach in the preparation of GO films. Raman spectroscopy analysis indicated the emergence of the G band, which significantly signifies the repair of lattice defects and confirms that the structural integrity of GO was restored after treatment. This phenomenon not only validates the effectiveness of the treatment but also provides important insights into the improvement of material properties. Additionally, conductive atomic force microscopy (C-AFM) was employed to characterize the local conductivity at the nanoscale, revealing a positive correlation between the effective removal of oxygen-containing functional groups and the increase in conductivity as the plasma treatment time was extended.

Yang et al. [95] proposed an innovative method that combines methane and argon to rapidly reduce GO at room temperature using inductively coupled plasma (ICP). Its reduction mechanism is shown in Figure 7. The results demonstrated that when the volume ratio of methane to argon was 2:1 and the treatment time was 5 min, the C/O ratio of the resulting reduced GO significantly increased to 10.6, while the conductivity reached 264.5 S/m. This outcome illustrates the method’s effectiveness in significantly reducing oxygen content while greatly enhancing the electrical properties of the material. Compared to traditional methods, this approach is straightforward and does not require high-temperature treatment. The reduction process at room temperature reduces energy consumption and minimizes the risk of compromising thermal stability. The application of ICP technology effectively enhances the reactivity of gas molecules, improving the efficiency of the reduction reaction.

### 3.4. Nitrogen and Ammonia

Nitrogen plasma is a commonly used method for the reduction and doping of GO [96,97]. Doping with heteroatoms can effectively alter the structure and properties of carbon-based materials [98,99]. When graphene sheets are doped with nitrogen atoms, three common bonding configurations arise within the lattice: pyridinic N, graphitic N, and pyrrolic N. Compared to other methods, employing nitrogen and ammonia plasma allows for a higher concentration of nitrogen atoms. In the case of GO, plasma treatment not only promotes N atom doping but also facilitates concurrent deoxygenation and reduction. Compared to inert gases such as argon or helium, nitrogen gas exhibits superior reduction efficiency when used as a working gas [100]. In plasma, nitrogen is excited to produce high-energy reactive species, such as nitrogen atoms and N_2_* excited state molecules. These species can chemically react with the oxygen-containing functional groups on GO, converting them into volatile products (such as NO or NO_2_), thereby more effectively removing the oxygen groups [101]. Additionally, nitrogen plasma treatment can enable nitrogen doping by forming pyridinic, pyrrolic, or graphitic nitrogen, which further increases the C/O ratio, significantly enhancing the material’s conductivity and chemical stability [102]. In contrast, inert gases primarily remove oxygen functional groups through physical bombardment, lacking the chemical reactivity needed for efficient reduction and doping. GO treated with nitrogen plasma typically achieves a C/O ratio of 7–8, whereas treatment with inert gases usually results in a C/O ratio of 4–6.

Santhosh et al. investigated the treatment of graphene nanowalls (CNWs) with different nitrogen-containing plasmas [103]. They found that N-CNWs treated with nitrogen plasma exhibited a higher concentration of graphitic N, whereas those treated with ammonia plasma exhibited a higher concentration of pyridinic N. Therefore, selecting an appropriate nitrogen-containing plasma environment allows for effective control over the nitrogen configuration. Akada et al. used radiofrequency (RF) plasma treatment in a nitrogen environment to reduce GO [104]. The treatment was carried out at a power of 50 W and a treatment time of 20 min. Through X-ray photoelectron spectroscopy (XPS), they discovered that the nitrogen plasma treatment not only reduced oxygen-containing groups but also simultaneously doped the GO with nitrogen, as shown in Figure 8. This dual effect of reduction and doping can be performed at room temperature, preventing the desorption of dopants and functional groups. The nitrogen doping led to an increase in the nitrogen content in GO, as confirmed by the XPS spectra. This study demonstrated the surface modification capabilities of plasma treatment, emphasizing its ability to control both reduction and doping processes effectively. The authors proposed that this approach could lead to improvements in the performance of devices such as supercapacitors and sensors.

Chemical vapor deposition (CVD) is a highly attractive technique for heteroatom-doped graphene [105,106,107,108]. Typically, graphene is grown on a catalytic substrate via CVD, necessitating a complex transfer process to etch away the metal catalyst and transfer graphene onto the desired surface. However, this transfer process can lead to the generation of defects in graphene and a reduction in carrier mobility and conductivity, making damage almost unavoidable. Meskinis et al. achieved the synthesis of catalysis-free and transfer-free graphene while doping with nitrogen using microwave plasma-enhanced CVD [109]. By varying the nitrogen flow rate, they explored the structure and chemical properties of nitrogen-doped graphene, leading to the preparation of nitrogen-doped multilayer graphene structures.

## 4. Applications for RGO in Energy Storage Devices

Graphene’s exceptional mechanical and physical properties make it an ideal material for energy storage applications [110,111,112,113,114]. For lithium-ion batteries, rGO-based electrodes significantly outperform traditional graphite anodes. While graphite provides a stable specific charge capacity of 372 mAh/g, rGO can achieve capacities exceeding 1500 mAh/g under optimized conditions. This improvement is attributed to rGO’s larger specific surface area, superior electrical conductivity, and structural flexibility. However, challenges like aggregation due to π-π interactions must be mitigated to fully utilize these properties, and rGO also shows remarkable performance enhancements over conventional carbon materials. [115] For example, rGO-based supercapacitors achieve a specific capacitance of up to 1249 F/g at a current density of 1 A/g, compared to the lower capacitance observed in simple carbon-based EDLCs. Moreover, composite materials like rGO@Co_3_O_4_ demonstrate improved energy densities, reaching up to 23.6 Wh/kg, which is significantly higher than conventional carbon electrodes. These enhancements are crucial for wearable devices and electric vehicle applications, where high energy and power densities are required [116,117,118].

### 4.1. Materials and Methods

Traditionally, graphite has been the anode material used in commercial lithium-ion batteries. Its layered structure allows for the intercalation of lithium ions and provides a stable charge capacity. However, the theoretical specific charge capacity of graphite is limited to only 372 mAh/g, and its charge–discharge process is relatively slow compared to actual requirements [119,120]. Additionally, during the battery charging process, the formation of metallic lithium dendrites (lithium plating) can occur. In contrast, graphene offers a larger specific surface area and superior electrical conductivity, chemical stability, mechanical stability, and flexibility compared to graphite. However, the limitations of graphene primarily arise from its tendency to aggregate due to van der Waals forces and π-π interactions between graphene layers. This aggregation reduces its active surface area and hinders the efficient transport of lithium ions, which must reside on the graphene layers. Therefore, alternative methods are necessary to enhance performance.

Phosphorus, known for its high theoretical capacity and low cost, is a promising anode material. However, its significant volume expansion and poor conductivity hinder its practical application. The hybridization of transition metal phosphides with layered rGO presents an ideal structure to overcome these challenges. Chen et al. synthesized a layered Cu3P@rGO nanostructure using a mild, low-temperature plasma phosphorylation process [121], as illustrated in Figure 9. This electrode demonstrated an impressive specific capacity of 2330 mAh/g and retained over 80% of its initial Coulombic efficiency after 183 cycles at a current density of 500 mA/g.

Furthermore, transition metal oxides exhibit high specific capacity and energy density, which contribute to relatively high battery energy density. However, during repeated electrochemical cycling, the electrodes experience significant pulverization due to drastic volume changes of up to 200% and slow conversion reaction kinetics. Macroscopically, this leads to poor cycling stability and reduced lithium storage capacity. To further enhance performance, rGO is often combined with other functional materials. For example, Cu_3_P@rGO and Fe_3_O₄/rGO composites exhibit a specific capacity of over 2000 mAh/g and excellent cycling stability. These improvements make rGO a strong candidate for next-generation high-performance lithium-ion batteries. Zhou et al. [122] controlled the in situ nucleation and growth of β-FeOOH on GO (β-FeOOH/GO) and employed DBD hydrogen plasma to fabricate Fe_3_O_4_/rGO composites, as illustrated in Figure 10. These composites demonstrated high reversible capacity, good rate capability, and excellent stability, maintaining a capacity of 890 mAh g^−1^ over 100 cycles at a current density of 500 mA g^−1^.

Another approach is to utilize this material in a three-dimensional configuration, where the porous structure enhances electrolyte penetration, improves electrode wetting, and increases the contact area between the electrolyte and the electrode, thereby reducing the diffusion distance of Li^+^ ions. Abdollahi et al. [123] employed plasma-enhanced chemical vapor deposition to fabricate vertically aligned carbon nanotubes (VA-CNTs) on independently porous activated reduced GO paper, as illustrated in Figure 11. Compared to GO, VA-CNTs demonstrated higher specific capacitance and superior rate performance. Furthermore, the low-defect sp^2^ carbon hybridization of the structure contributed to its exceptional cycling stability, making it suitable for high-performance lithium-ion battery anodes.

### 4.2. Supercapacitor

Advanced supercapacitors featuring high energy and power density, exceptional reversibility, and extended cycle life have become a focal point of interest [124].

Based on the charge–discharge mechanisms of the electrodes, supercapacitors are generally categorized into two types: (1) electrochemical double-layer capacitors (EDLCs) and (2) pseudocapacitors. EDLCs operate on the principle of adsorption and desorption of electrolyte ions at the electrode/electrolyte interface, while pseudocapacitors store charge through redox reactions that involve electron transfer processes. However, current electrode materials encounter significant limitations and drawbacks. For example, the materials used in EDLCs, which are primarily carbon-based, exhibit low specific capacitance. In contrast, pseudocapacitor materials, such as transition metal compounds and conductive polymers, often suffer from poor cycling stability.

With the increasing demand for supercapacitor applications, simple graphene materials and transition metal oxide electrodes are no longer sufficient to meet the requirements. There is an urgent need to develop new electrode materials for supercapacitors that offer high energy and power densities, as well as excellent cycling stability. Consequently, the fabrication of composite materials that combine graphene with transition metal oxides has become a focal point of research. Zhang et al. [125] synthesized rGO@Co_3_O_4_ composite materials using liquid-phase plasma electrolysis, as illustrated in Figure 12. Electrochemical tests demonstrated that this composite material exhibited a higher specific capacitance and better long-term cycling stability compared to graphene alone. Specifically, the composite achieved a specific capacitance of 1249.0 F/g at a current density of 1 A/g, with a capacitance retention rate of 89.7% after 10,000 cycles. Supercapacitors assembled with rGO and rGO@Co_3_O_4_ composites demonstrated an energy density of 23.6 Wh/kg at a power density of 0.4 kW/kg and maintained an energy density of 16.0 Wh/kg at a power density of 6.8 kW/kg. Moreover, the capacitance retention rate at a current density of 5 A/g was 88.2%. Long et al. [126] synthesized Co_3_O_4_/graphene nanocomposites by uniformly embedding Co_3_O_4_ nanoparticles into graphene nanosheets using a plasma-assisted processing method. This nanocomposite exhibited a high specific capacity of 1368 mAh/g at a current density of 125 mA/g, along with excellent cyclability and rate capability.

Another method to modify the electronic properties of graphene is through the doping of heteroatoms, such as boron (B), nitrogen (N), phosphorus (P), and sulfur (S), into the graphene structure using chemical or physical techniques. This process introduces either n-type or p-type conductivity and creates defects in the vicinity due to the uneven distribution of charges. These defects facilitate charge transfer between adjacent carbon atoms, thereby enhancing the electrochemical performance of doped graphene materials. Li et al. [91] developed boron-doped reduced graphene oxide (B-rGO) by subjecting a mixture of GO and boric acid to DBD plasma treatment. This approach enabled the efficient and effective doping of boron atoms into the graphene structure, achieving a concentration of approximately 1.4 atomic percent. The specific capacitance of the doped graphene material was significantly higher than that of pristine graphene, demonstrating substantial potential for the large-scale production of heteroatom-doped graphene.

Furthermore, nitrogen-doped graphene typically exhibits a larger surface area. Nitrogen functionalization is known to alter the donor/acceptor characteristics of graphene, and n-doping can introduce numerous structural defects. These defects can serve as active sites for electron storage, thereby improving capacitance performance. Jeong et al. [127] synthesized nitrogen-doped graphene (as depicted in Figure 13) for high-performance supercapacitors and investigated the significance of nitrogen doping sites on the basal plane. Their study employed a straightforward plasma process to fabricate nitrogen-doped graphene, resulting in a capacitance approximately four times higher than that of pristine graphene, while preserving other fundamental and beneficial properties of supercapacitors. The high performance of rGO is primarily attributed to its large specific surface area and conductive network, while doping techniques (such as nitrogen doping) further enhance its ion insertion and storage capabilities. Future research should focus on optimizing composite fabrication and doping methods to further improve specific capacitance and cycling stability.

## 5. Conclusions

To gain a deeper understanding of the plasma preparation process for rGO, this study first analyzes the structure of GO and introduces traditional reduction methods, such as chemical and thermal reduction. Although these methods are well established, they have limitations, including the use of toxic reducing agents, product impurities, and high costs. Plasma reduction, however, effectively removes oxygen functional groups, eliminates impurities depending on the working gas, repairs carbon surface defects, and allows for tailored doping, offering advantages like lower energy requirements and precise control over the reduction and doping processes.

The study also highlights the role of gas environments in plasma reduction, focusing on three aspects: (I) enhancing conductivity by increasing the C/O ratio, (II) simultaneous elemental doping, particularly nitrogen insertion, and (III) restoring the graphite network through sp^2^ carbon hybridization. The partial restoration of the sp^2^ network in rGO significantly improves its conductivity, ranging from 10 to 500 S·cm^−1^, compared to the electrically insulating GO. Nitrogen-containing gases, like N_2_ or NH_3_, further enhance rGO’s electrochemical activity and stability by doping nitrogen into the carbon lattice, which benefits energy storage applications.

Plasma-prepared rGO shows significant promise in energy storage devices such as lithium-ion batteries and supercapacitors, offering superior electrochemical performance compared to traditional graphite anodes, with higher reversible capacities (~1500 mAh/g) and faster charge–discharge rates. However, challenges such as low purity, interlayer stacking, and high liquid absorption remain. Future developments should focus on improving rGO purity, reducing stacking, and enhancing performance through multi-element doping or composite strategies. Plasma reduction has great potential for large-scale applications in energy storage, such as grid-level storage and EV fast charging stations, with further research needed to optimize purity, scalability, and integration with advanced materials.

## Figures and Tables

**Figure 1 nanomaterials-14-01922-f001:**
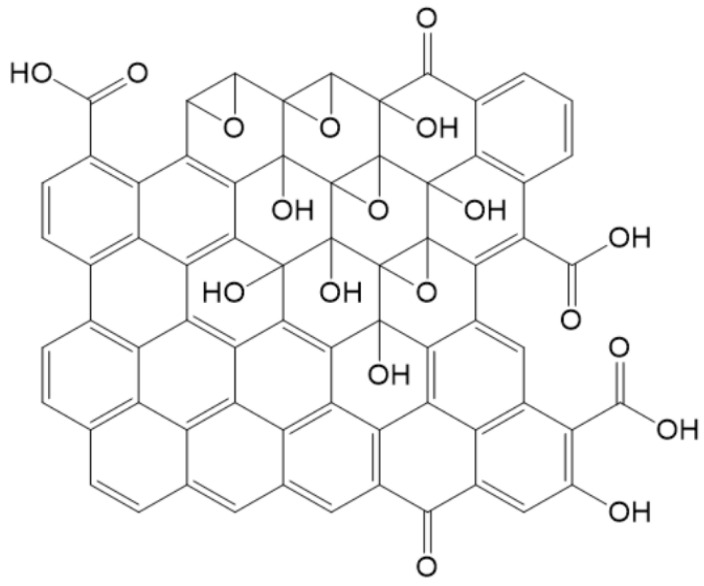
The LK model of GO [42].

**Figure 2 nanomaterials-14-01922-f002:**
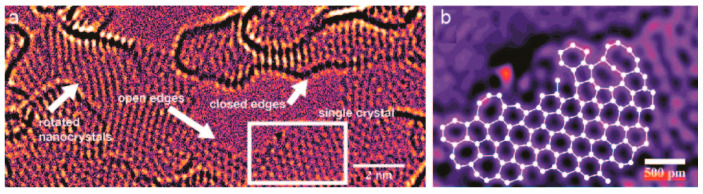
TEM image of GO [31]. (**a**) The atomic structure of GO; (**b**) the chemical bonding structure at higher resolution.

**Figure 3 nanomaterials-14-01922-f003:**
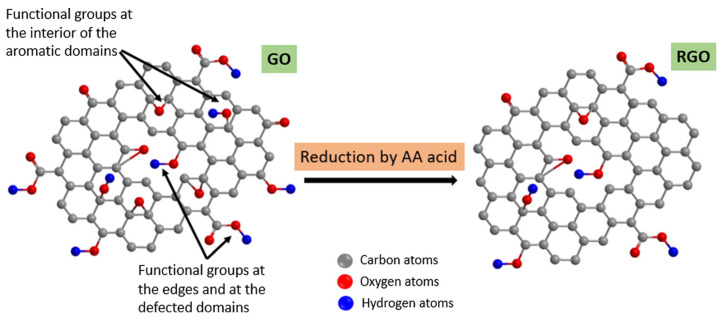
Schematic representation of the oxygen functionalities present in GO and RGO [56].

**Figure 4 nanomaterials-14-01922-f004:**
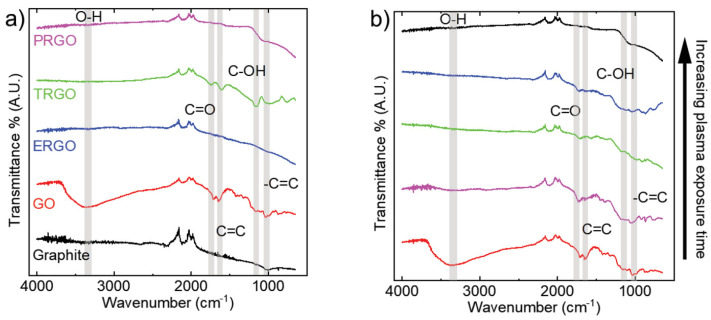
FTIR spectra of (**a**) plasma-reduced GO (PRGO), thermally reduced GO (TRGO), and electrochemically reduced GO (ERGO), graphite, and unreduced GO; (**b**) PRGO as a function of plasma treatment time (30, 60, 90, and 120 min) [84].

**Figure 5 nanomaterials-14-01922-f005:**
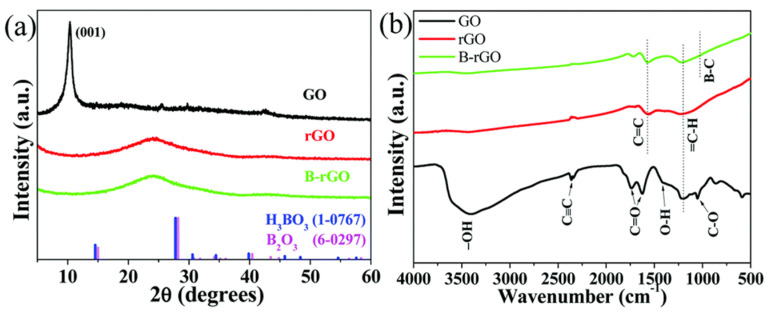
(**a**) XRD patterns and (**b**) FT-IR spectra of the prepared samples [90].

**Figure 6 nanomaterials-14-01922-f006:**
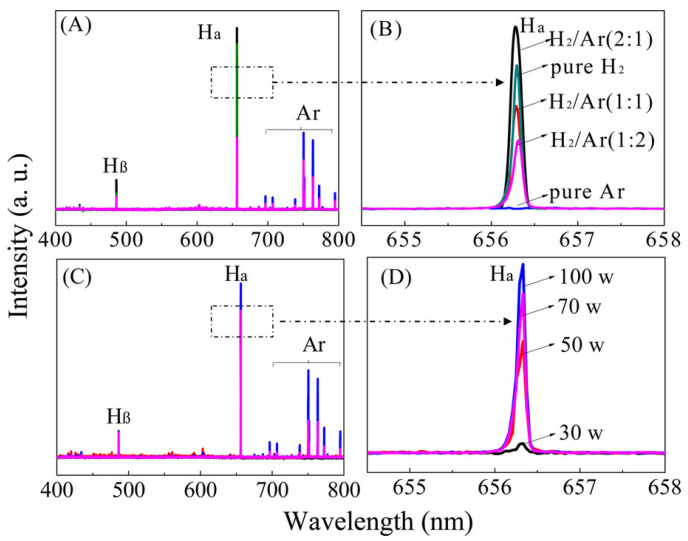
Emission spectra of H_2_/Ar plasma as a function of the ratio of H_2_ to Ar (**A**,**B**) and discharge power (**C**,**D**). Lines corresponding to Ar excited states and atomic hydrogen (Hα and Hβ) are indicated [91].

**Figure 7 nanomaterials-14-01922-f007:**
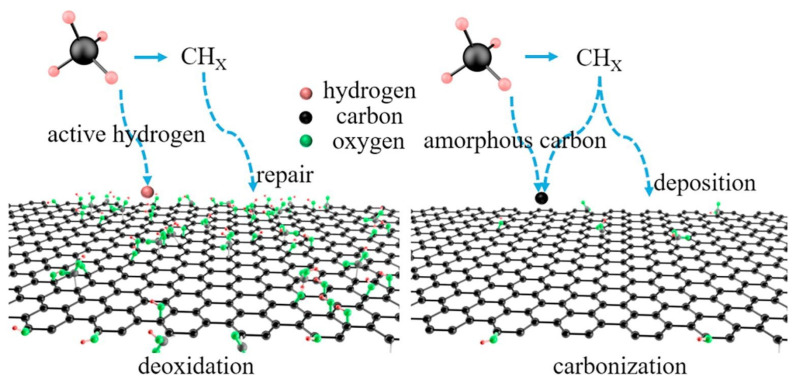
A schematic diagram of the proposed reduction mechanism of CH_4_/Ar plasma treatment [95].

**Figure 8 nanomaterials-14-01922-f008:**
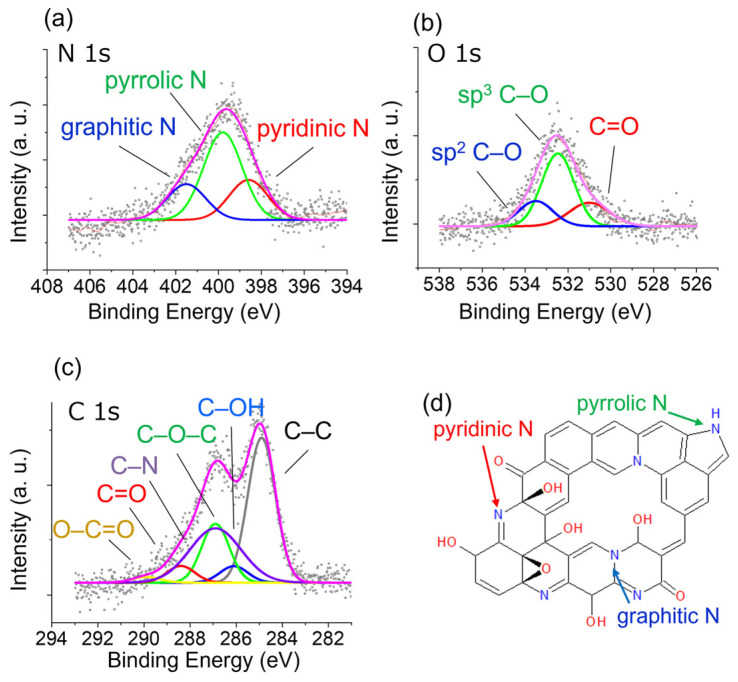
(**a**) N 1s, (**b**) O 1s, and (**c**) C 1s XPS spectra of GO-treated N_2_ plasma for 60 min. (**d**) Schematic of N-doped GO structure [104].

**Figure 9 nanomaterials-14-01922-f009:**
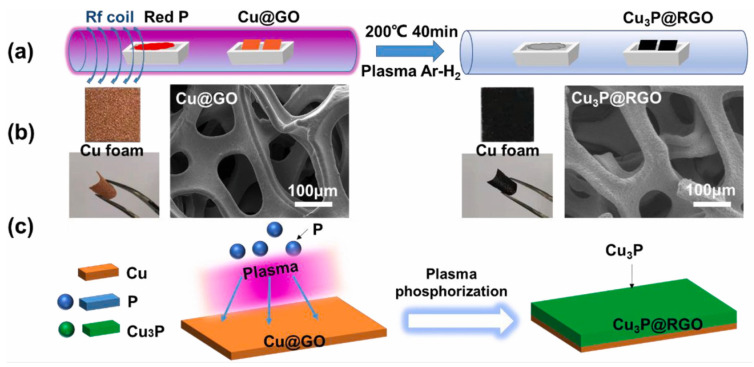
Schematic diagram of the synthesis of a Cu3P@RGO electrode: (**a**) experimental process, (**b**) electron image, and (**c**) mechanism of plasma phosphorylation [121].

**Figure 10 nanomaterials-14-01922-f010:**
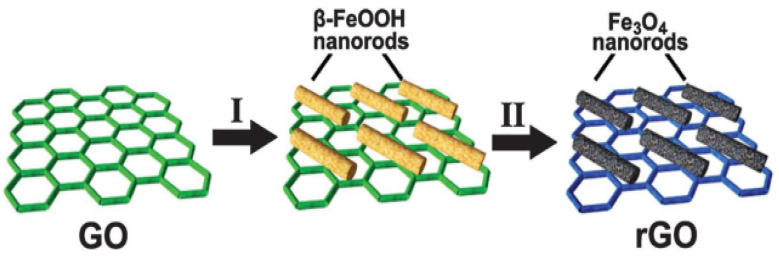
Schematic diagram of GO modified with mesoporous Fe_3_O_4_ nanorods [122].

**Figure 11 nanomaterials-14-01922-f011:**
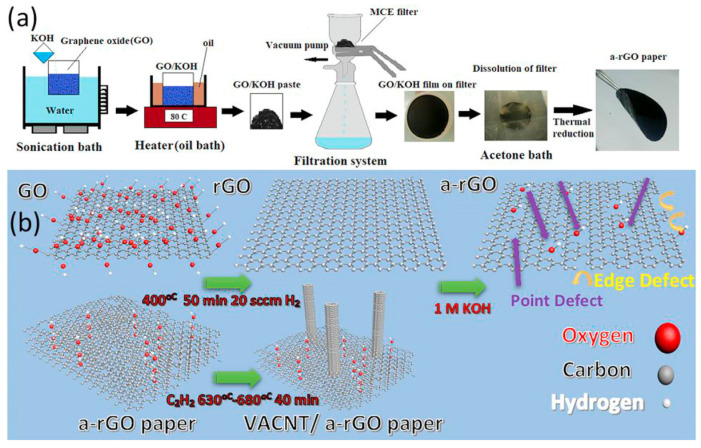
Schematic diagram of the preparation of VACNT/a-rGO hybrid nanostructures [123]. (**a**) Diagram showing the fabrication process of free-standing a-rGO paper, (**b**) The atomic scale preparation of VACNT/a-rGO and the effect of each step on the atomic structure and surface properties.

**Figure 12 nanomaterials-14-01922-f012:**
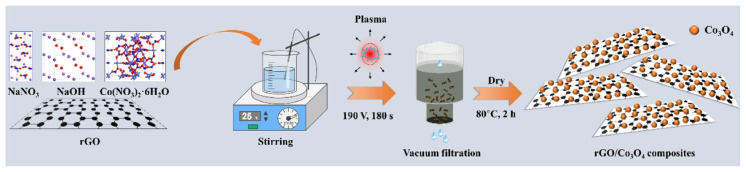
Schematic diagram of fabrication process for rGO@Co_3_O_4_ composite material [126].

**Figure 13 nanomaterials-14-01922-f013:**
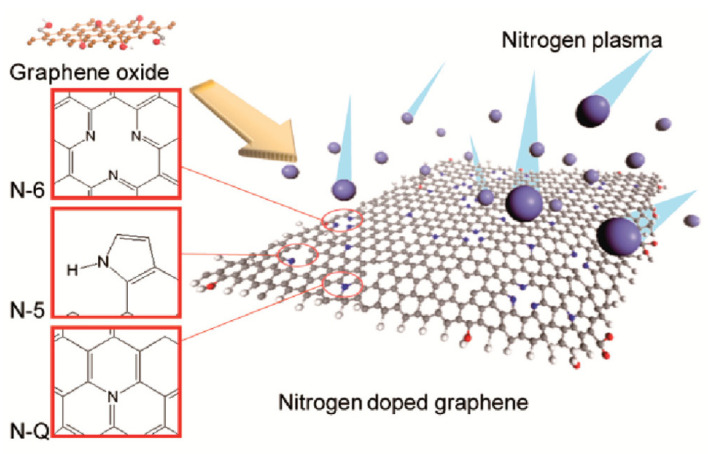
Schematic of plasma nitrogen doping [127].

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
