# Peer review of "Plasma-Assisted Preparation of Reduced Graphene Oxide and Its Applications in Energy Storage"

_nanomaterials, 2024, doi:10.3390/nano14231922_

Round 1

Reviewer 1 Report

Comments and Suggestions for Authors

This manuscript delves into recent advancements in the reduction of graphene oxide (GO), with a particular focus on plasma-assisted techniques. Although the paper is well-structured and offers compelling insights in its conclusion, major revisions are needed before it can be considered for publication.

1.      While the paper effectively highlights the benefits of various plasma environments, it lacks a detailed explanation of the chemical reaction mechanisms involved in plasma reduction. Please provide a more detailed explanation.

2.      Please provide a more detailed procedure for reducing graphene oxide. For example, in the introduction section, specify the precise temperature used for mild conditions. Additionally, including quantitative performance metrics and comparisons with traditional reduction methods would be beneficial.

3.      The manuscript would benefit from more detailed explanations of the figures throughout. For example, in figure 8, clarify the exact conditions for the Thermal and Ar plasma treatments, and explain whether the nitrogen plasma treatment simultaneously reduced and doped the GO with nitrogen.

4.      In section 3.4, "Nitrogen and Ammonia," the authors focus on the effects of nitrogen doping. However, readers may also be interested in understanding how the reduction ratio compares to that achieved with other inert gases. Please consider adding more detailed explanations on this comparison.

5.      In section 5, "Applications for RGO in Energy Storage Devices," the authors discuss the application of GO but provide limited detail on its specific properties, comparisons with conventional graphite, and the improvements achieved. Please consider elaborating on these aspects for better clarity.

6.      While the conclusions are intriguing, it would be even better if potential applications in actual energy storage systems or future directions for development were discussed.

7.      The authors should carefully proofread the manuscript concerning typos and grammar errors.

8.      Some figures have low image resolution. Please revise them.

Reviewer 2 Report

Comments and Suggestions for Authors

In this article, the authors have described the properties of RGO, their preparation methods, and their applications in energy storage devices. RGO is a highly crucial material because of its incredible properties, which make it essential for utilizing in various electrochemical energy storage devices such as batteries and supercapacitors. The authors have explored the applications of plasma-assisted methods for RGO preparation and their further utilization in this review article. It has also been observed that inert gas plasmas, such as Argon (Ar) and Helium (He), exhibit magnificent reduction efficiency, while mixed gases enable simultaneous impurity reduction. This review clearly explains the applications of plasma-assisted RGO in energy storage devices. Therefore, I recommend the acceptance of this article after minor revisions:

1. The title of the article should be improvised as the article is more focused on plasma-assisted RGO applications.

2.      Articles possess some typos.

3.      In the third section, “When solid surfaces are exposed to plasma, two simultaneous processes occur: (i) material deposition and (ii) ablation, which leads to material removal.” Explain more about these processes.

4.      The working of plasma-assisted reduction should be elaborated.

5.      Mention the full “PRGO, TRGO, ERGO” forms.

6.      More explanation on electrochemical reduction and its comparison is required.

7.      Include more challenges and future pathways.

Round 2

Reviewer 1 Report

Comments and Suggestions for Authors

no more comments